# Operando visualisation of battery chemistry in a sodium-ion battery by $^{23}$Na magnetic resonance imaging

Joshua M. Bray[1], Claire L. Doswell [1], Galina E. Pavlovskaya [2,3], Lin Chen [4], Brij Kishore [4], Heather Au[5], Hande Alptekin[5], Emma Kendrick [4], Maria-Magdalena Titirici [5], Thomas Meersmann[2,3] & Melanie M. Britton [1]✉

Sodium-ion batteries are a promising battery technology for their cost and sustainability. This has led to increasing interest in the development of new sodium-ion batteries and new analytical methods to non-invasively, directly visualise battery chemistry. Here we report operando $^1$H and $^{23}$Na nuclear magnetic resonance spectroscopy and imaging experiments to observe the speciation and distribution of sodium in the electrode and electrolyte during sodiation and desodiation of hard carbon in a sodium metal cell and a sodium-ion full-cell configuration. The evolution of the hard carbon sodiation and subsequent formation and evolution of sodium dendrites, upon over-sodiation of the hard carbon, are observed and mapped by $^{23}$Na nuclear magnetic resonance spectroscopy and imaging, and their three-dimensional microstructure visualised by $^1$H magnetic resonance imaging. We also observe, for the first time, the formation of metallic sodium species on hard carbon upon first charge (formation) in a full-cell configuration.

[1] School of Chemistry, University of Birmingham, Edgbaston, Birmingham B15 2TT, UK. [2] Sir Peter Mansfield Imaging Centre, School of Medicine, University of Nottingham, Nottingham NG7 2RD, UK. [3] NIHR Nottingham Biomedical Centre, Nottingham NG7 2RD, UK. [4] School of Metallurgy and Materials, University of Birmingham, Birmingham B15 2TT, UK. [5] Department of Chemical Engineering, Imperial College London, South Kensington Campus, SW7 2AZ London, UK. ✉email: m.m.britton@bham.ac.uk

The development of new battery technologies and electrochemistries is driven by ever-increasing demands of society, industry and the environment. Sodium-ion batteries (NIBs) offer advantages in cost and sustainability over current Li-ion batteries (LIBs), while still providing high energy density, which are particularly useful for short distance transportation and stationary storage[1,2]. However, many technological hurdles remain to be overcome before their widespread commercialisation. Although NIBs and LIBs face common lifetime and safety issues (including electrolyte decomposition, capacity fade and dendrite formation), NIBs also suffer from unique challenges associated with the development of appropriate anode materials and the identification of optimised electrolytes[3–5]. Although graphite is a widely used anode in LIBs, it has little capacity to store sodium, and therefore development of NIB anodes from high-capacity, low cost and renewable sources is an active area of research[6]. Primary candidate materials include non-graphitic carbons created from pyrolysis of organic and polymer precursors (hard carbons) or derived from aromatic starting materials (soft carbons)[5,7–9], with the main distinction being that hard carbons cannot be graphitised, even at temperatures exceeding 2000 °C. Both types of carbon are composed of graphene layers, but incorporate structural disorder via defects, non-parallel alignment of domains, variable interlayer $d$-spacing, as well as microstructure and porosity, each of which provides mechanisms for sodiation capacity[3,5]. With hard carbon anodes, the voltage-charge curves generally exhibit a characteristic low-potential plateau, which can account for as much as 50% of the capacity. Practical concerns with this plateau are, that at this voltage, so close to Na$^+$/Na, metallic Na plating may occur, although the mechanism by which Na enters the pores is still unknown at low potentials. Thus, there is a risk that dendrites may form, which presents the risk of short circuiting and explosion[10–12]. It also means increased sensitivity to polarisation and risk of electrolyte degradation[5]. Proposed mechanisms for sodiation in non-graphitic carbons include reversible processes, such as intercalation of Na$^+$ in graphene domains with large interlayer $d$-spacing or filling of nanopores[5]; as well as irreversible ones, such as the trapping of Na in carbon particles, which produces local and macroscopic expansion of the electrode material[3]. In theory, the most desirable anode for a NIB is Na metal, but as with LIBs, a major barrier to commercialisation is an even greater tendency to form dendrites.

Alongside increasing research activity into NIB materials, there is also a pressing demand for non-invasive imaging techniques to probe batteries in operando, enabling better understanding and control of battery performance and degradation. The demand for these new operando analytical techniques is particularly relevant for NIBs, where there is a critical need to identify optimised standard electrolytes and improved electrode materials, and to better understand the factors controlling the composition and stability of the solid-electrolyte interphase (SEI) and the formation of dendrites. A major challenge for the characterisation of NIB storage mechanisms is that certain states of Na within the electrodes are metastable and materials in the cell may be highly sensitive to changes in environmental conditions when a battery is dismantled for post mortem analysis. Hence, in situ techniques are required. Previously, in situ $^{23}$Na nuclear magnetic resonance (NMR) spectroscopy has been able to quantify the deposition of microstructural Na on metallic anodes[10] and infer the chemical environment—and thus storage mechanisms—in non-graphitic hard carbon anodes[13]. $^{23}$Na NMR shows a large chemical shift difference for the metallic, compared to solvated (diamagnetic), sodium in the electrolyte, which arises from the Knight shift of the NMR signal[10]. In addition, a shift in the $^{23}$Na NMR signal for solvated Na in the electrolyte has been observed in the pores of

microporous carbon electrodes, compared to that within the porous glass separator, which is due to ring current effects from the pore walls in the carbon, giving rise to a nucleus-independent chemical shift (NICS)[14,15]. Operando solid-state NMR studies of hard carbons[13] have shown $^{23}$Na NMR signals attributed to both Na metal and Na$^+$ in the electrolyte, but have also shown an additional peak at −40 ppm, which shifts to +600 ppm during the low-potential (0.1 V) plateau. This additional peak has been attributed to Na$^+$ adsorbed at defect sites in the carbon, which subsequently moves downfield due to intercalation and "pooling" of sodium between graphene layers[13]. Comparison of these spectra with ex situ magic angle spinning (MAS) NMR has revealed extreme sensitivity to trace air/moisture, and the peak associated with Na in the carbon has been found to disappear over a few hours[13], which explains why it has been observed in some studies[16], but not others[17]. Such observations, in particular, underline the need for in situ and operando methods, which are essential for accurately identifying sodiation mechanisms.

In recent years, magnetic resonance imaging (MRI) has emerged as a powerful operando technique to study electrochemical devices, spurred on by advances that mitigate imaging artifacts associated with metallic electrodes[11,18–21]. Although MRI has lower spatial resolution than X-ray or electron microscopies, it is able probe both electrode and electrolyte environments, providing a more holistic view of metallic counter electrode (half-cell) and full-cell battery configurations[22]. Moreover, the synergy between NMR spectroscopy and imaging modalities enables information at a molecular level to be correlated with spatially heterogeneous properties at a mesoscopic ($10^{-7}$–$10^{-5}$ m) and macroscopic ($10^{-3}$–$10^{-2}$ m) level by MRI[19]. In LIBs, electroactive species containing NMR active $^6$Li or $^7$Li have been imaged directly by MRI[11,12,23–25]. Where electroactive species do not contain NMR active nuclei, it is has been found that these can be imaged indirectly using $^1$H or $^{19}$F MRI of the electrolyte[26] or counter ions—an approach that has been used to image battery chemistry in LIBs and Zn-air batteries[12,18,24]. Although $^{23}$Na NMR spectroscopy has been employed to study NIBs[10], two-dimensional (2D) MRI is more technically demanding for $^{23}$Na, because the standard (spin-warp) acquisition schemes used for $^1$H or $^7$Li MRI have to be changed to accommodate the significantly faster transverse relaxation of the $^{23}$Na nucleus, caused by its strong nuclear electric quadrupole interactions.

In this paper, we report, to our knowledge for the first time, operando $^1$H and $^{23}$Na NMR and MRI experiments observing the speciation and distribution of Na in the electrode and electrolyte, during sodiation and desodiation of hard carbon, in a sodium metal cell and a sodium-ion full-cell. The formation and evolution of sodium dendrites are monitored by $^{23}$Na NMR spectroscopy and imaging, and their three-dimensional microstructure visualised by $^1$H MRI. Furthermore, we observe, to our knowledge for the first time, the formation of metallic sodium species on hard carbon upon first charge (formation) in a full-cell configuration.

## Results

**Pristine sodium metal cell.** A schematic diagram of the sodium metal cell is shown in Fig. 1a, along with $^{23}$Na NMR spectra, 2D images and 1D profiles for a pristine cell. The $^{23}$Na NMR spectra (Fig. 1b), shows a peak around 0 ppm, arising from solvated sodium species in the EC/DMC electrolyte, and a peak at 1131 ppm, arising from metallic sodium in the CE, which is at a significantly higher frequency due to the Knight shift of the NMR signal[10]. A deconvolution of the peak at 0 ppm, reveals two peaks at 0 and −4 ppm, which has been assigned to Na species in the electrolyte, within the separator (0 ppm), and excess electrolyte,

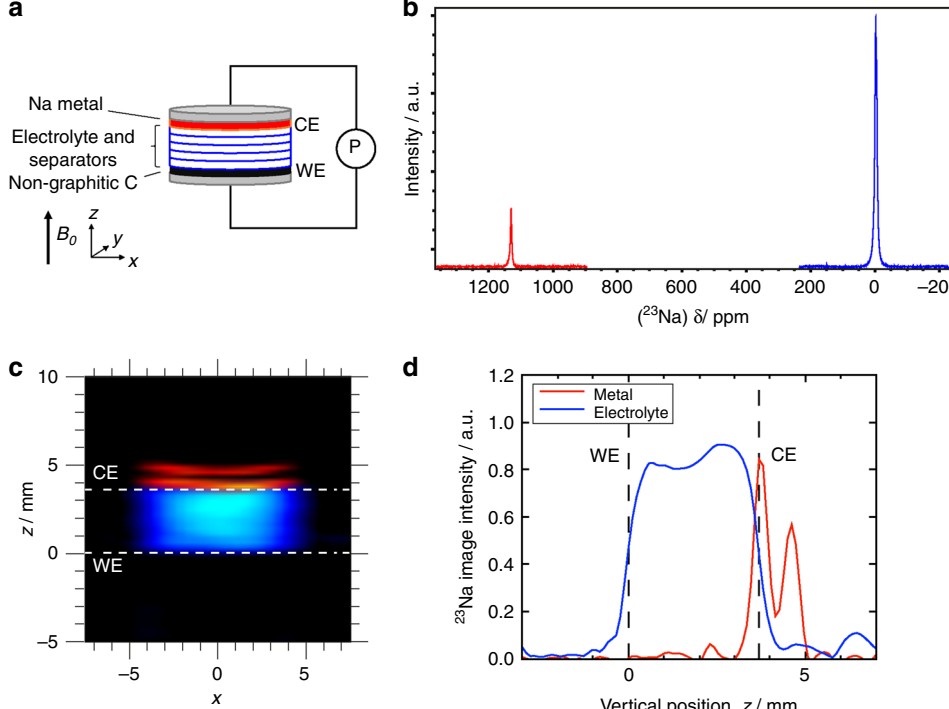

**Fig. 1 $^{23}$Na NMR signal in a pristine sodium metal cell. a** Schematic diagram of the sodium metal cell with respect to the imaging coordinate system and the $\boldsymbol{B}_0$ magnetic field direction. The working electrode (WE) and counter electrode (CE) are connected to a potentiostat (*P*). **b** $^{23}$Na NMR spectra, acquired separately, for Na in the dielectric environment of the electrolyte (blue spectrum) and for the metallic Na (red spectrum). **c** 2D $^{23}$Na MRI of metallic (red) and dielectric (blue) sodium in the cell. The dashed lines indicate the positions of the sodium metal CE and carbon WE. **d** 1D $^{23}$Na MR profile of metallic (red) and dielectric (blue) sodium in the cell. This cell was assembled with five separators.

which has been squeezed outside the cell but is contained within the Swagelok system (−4 ppm). The intensity of the metallic signal, at 1131 ppm, is smaller than the solvated Na species, because of the skin-depth penetration of radiofrequency (RF) radiation[27] into the Na metal electrode, which is only 11 μm at 9.4 T. Hence, the NMR signal intensity is a function of the surface area only (and not the volume of $^{23}$Na present). 2D $^{23}$Na MR images of metallic (red) and solvated (blue) sodium are overlaid in Fig. 1c. A variation in signal intensity across the electrolyte in the *x*-direction can be observed, where the signal increases from the sides to the centre, which is due to the 3D cylindrical geometry of the cell projected onto a 2D plane. The Na metal counter electrode (CE) is above the separator stack, but only signal from the top and bottom surfaces of the Na electrode are observed, at positions $z \approx 4$ and 5 mm, due to the RF penetration[27]. Lastly, 1D $^{23}$Na MR profiles for both solvated and metallic sodium are shown in Fig. 1d. Increased signal intensity at the bottom and top faces of the Na metal CE is observed, as seen in the 2D images. The signal intensity is greater for the bottom face, than the top, which is expected because the top face is in contact with the aluminium current collector, which provides partial RF shielding there. The upper and lower extremities of the electrolyte signal correspond to the interfaces between the electrodes and electrolyte. These interfaces are not sharply defined, predominantly, because of blurring associated with the linewidth of the $^{23}$Na signal (≳300 Hz), as well as imperfections in the alignment and smoothness of the electrodes. Small amounts of electrolyte signal on the top of the current collector ($z > 5$ mm) can be observed in the 1D profiles (Fig. 1d), which are due to electrolyte that has been squeezed out of the separator stack. It is this signal that is believed to be associated with the signal observed at −4 ppm in the $^{23}$Na spectrum.

**Charge/discharge cycle in a sodium metal cell**. Figure 2 shows $^{23}$Na NMR spectra, 2D images and 1D profiles for a cell, which underwent a single discharge/charge (30 mA g$^{-1}$) cycle outside the magnet. The corresponding potential vs. capacity profile is shown in Supplementary Fig. 2, which exhibits the typical slope-plateau profile characteristic of hard carbons. The capacity is significantly less than that achieved in a traditional coin cell setup, which is due to the high resistivity of the cell configuration, and hence the lower voltage limit is reached before full sodiation occurs. This result, however, does give us a good indication of the different sodium speciation within or on the hard carbon, and also gives us the ability to study the morphology of sodium plating. The first cycle Coulombic efficiency is 10%, indicating that a the majority of the sodium transferred in the first cycle has either gone to form the electrolyte interface on the carbon or remains trapped in the carbon after charging. Due to the limited reversible capacity it is likely that we observed mostly the sodium speciation at the interface in this case. The peak for the metallic sodium (1131 ppm) remains unchanged, but there are now four peaks observed for the sodium in the electrolyte, at chemical shifts of 1, 0, −5 and −7 ppm (Supplementary Fig. 4 and Supplementary Table 3).The two peaks at 0 and −5 ppm have chemical shifts, peak intensities and line widths consistent with those observed in the pristine cell, for electrolytic sodium within the separator and excess 'free' electrolyte outside the cell (Fig. 1 and Supplementary Fig. 4). The additional peak at −7 ppm appears in the chemical shift range expected for sodium within the microporous carbon electrode[15], shifted by ring current effects. There are two stages of sodiation in hard carbon, the first follows a sloping voltage profile and is thought to be intercalation between the turbostatic graphene layers, the second is a voltage plateau near to 0 V vs. Na/Na$^+$, which is thought to be metal-like sodium

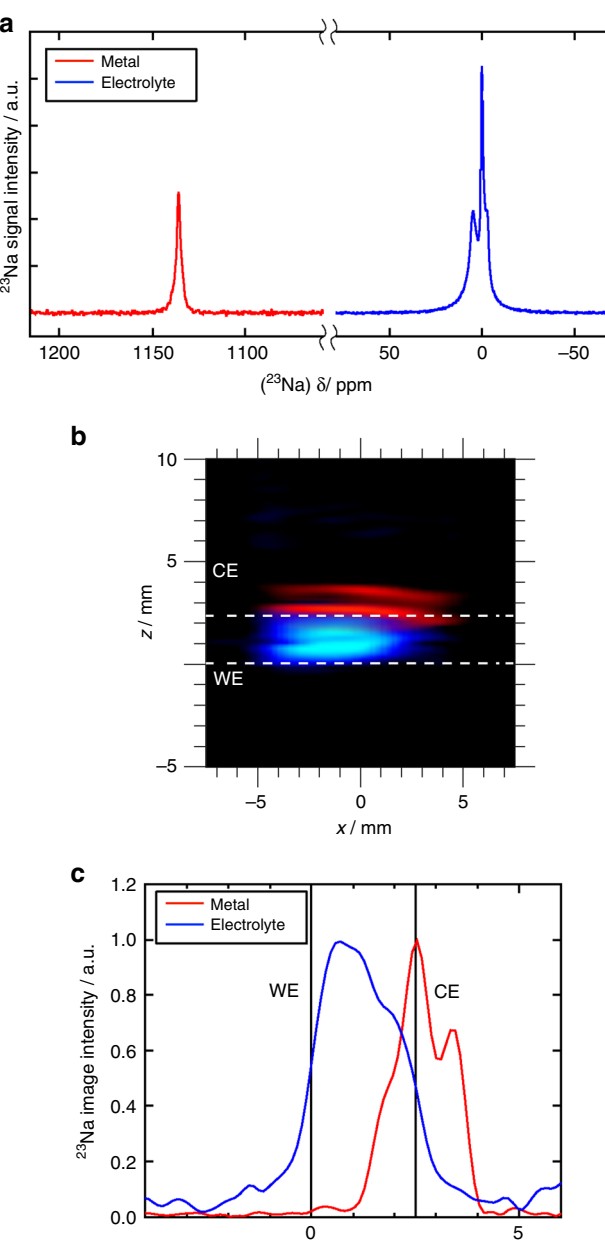

**Fig. 2 $^{23}$Na NMR signal after a single charge/discharge (30 mA g$^{-1}$) cycle. a** $^{23}$Na NMR spectra for solvated Na in the electrolyte (blue spectrum) and for metallic Na (red spectrum). **b** 2D $^{23}$Na MRI of metallic (red) and dielectric (blue) sodium in the cell. **c** 1D $^{23}$Na MR profile of sodium in metallic and electrolytic environments in the cell. This is cell was assembled with two separators.

the separators, indicating the formation of additional diamagnetic species, through degradation of the electrolyte, possibly on the surface of the electrode and forming part of the solid-electrolyte interphase (SEI)[13]. The presence of similar additional peaks was not observed previously during galvanostatic cycling in the in situ $^{23}$Na NMR study of a Na symmetrical cell with Na bis(tri-fluoromethylsulfonyl)imide (NaTFSI) in propylene carbonate (PC) electrolyte[10], but has been observed in ex situ $^{23}$Na NMR measurements of NIB hard carbon electrodes[28] with 1 M NaPF$_6$/ PC and fluoroethylene carbonate (FEC, 2 wt%) as an additive. Moreover, the appearance of a similar peak, believed to arise from species within the glass fibre separator, has been observed for solvated lithium species in an EC/DMC electrolyte of a lithium-ion battery following charge cycling, which was assigned to the formation of lithiated degradation products in the electrolyte[29]. In the sodium metal cell investigated in this work, the presence of the first additional sodium peak at −7 ppm is believed to be associated with the trapping of sodium within the carbon WE (see Supplementary Fig. 4 and Supplementary Table 3), whereas the second electrolytic peak at 1 ppm is attributed to the formation of degradation products, possibly part of the SEI layer. Also, the $^{23}$Na NMR spectra confirm that the sodium is inserted into the carbon electrode, rather than plated, as no additional metallic sodium signal is observed. A second charge cycle was performed on the cell, at a specific current of 20 mA g$^{-1}$ inside the magnet, but no significant changes were observed in the subsequent spectra or images (Supplementary Fig. 5).

**Galvanostatic plating in a sodium metal cell**. The promotion of galvanostatic plating was investigated following discharge at a specific current of 150 mA g$^{-1}$. Under these conditions, diffusion or intercalation of the sodium through the electrode is kinetically unfavourable and plating is expected to occur preferentially to insertion, which is consistent with the $^{23}$Na NMR spectra, 2D images and 1D profiles shown in Fig. 3. In the $^{23}$Na spectrum of the metallic sodium (Fig. 3a), an additional two peaks appear at 1141 and 1129 ppm, indicating the development of new Na metal environments. We attribute the additional peak at 1129 ppm to sodium that has been deposited at an angle close to 90°, with respect to the $B_0$ field[10], and hence parallel to the electrodes. Alternatively, it may arise from changes in surface morphology of the sodium counter electrode, as sodium is removed from the surface of the metal during the discharge. The peak at the higher field (1141 ppm), is expected to arise from sodium metal deposited at an angle closer to 0°, such as would be expected by the formation of dendritic Na microstructures, which would grow normal to the electrodes (and parallel to $B_0$). This second, higher intensity, peak we associate with the new metallic sodium environment observed on the WE in the 2D image (Fig. 3b) and 1D profile (Fig. 3c), suggesting the formation of sodium dendrites here. Similar structures have been observed spectroscopically in a NIB[10], as well as visualised by $^7$Li MRI in a LIB[11,30].

In addition to a change in the signal for the metallic sodium, there is also a change in the intensity of the peaks for solvated sodium, where a decrease is observed for the peak at 0 ppm (Fig. 3a and Supplementary Fig. 6a). The resonance at −5 ppm remains unchanged, as expected, as it is outside of the cell and will not participate in the electrochemistry within the cell. The signal at −7 ppm also appears to remain relatively unaffected during this process, as is expected because of the limited amount of sodium inserted into the carbon. However, the contribution from the broad peak at 1 ppm is approximately halved (Supplementary Fig. 6a and Supplementary Table 4), suggesting it is affected by the formation of the metallic sodium microstructures on the WE.

precipitating at the vertices of the layers, or pooling of the sodium. However, in this case the full sodiation was not observed and although both mechanisms have been observed to occur simultaneously, the very low capacity and only the sloping voltage profile suggests that in this case it is unlikely that the second sodiation species was observed, and only limited occurrence of the first sodiation species is likely due to the low specific capa-city[13,16]. The fourth peak, at a chemical shift (1 ppm) nearest to the electrolytic sodium in the separator, has a much broader linewidth (1900 Hz) compared to that of the peak at 0 ppm (450 Hz). This suggests these sodium species are less mobile, but in a comparable chemical environment to the electrolyte within

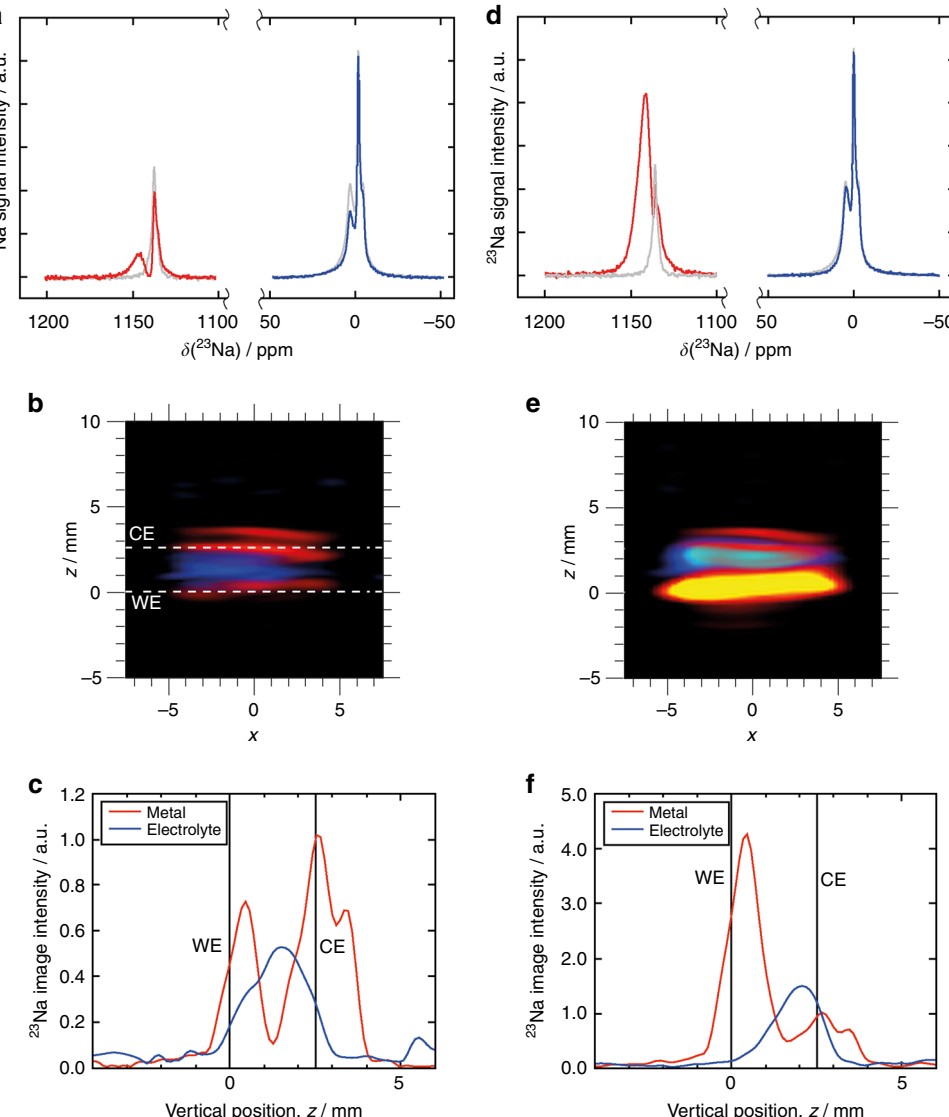

**Fig. 3 Evolution of $^{23}$Na NMR signal at the start and after continued galvanostatic plating. a**, **d** $^{23}$Na NMR spectra for solvated Na in the electrolyte (blue line) and for metallic Na (red line), after discharge at a specific current of 150 mA g$^{-1}$ (**a**) and after discharge at a specific current of 600 mA g$^{-1}$ (**d**). The corresponding spectra for the cell following a single charge cycle (Fig. 2a) are included for comparison as grey lines. **b**, **e** 2D $^{23}$Na MRI of metallic (red) and dielectric (blue) sodium in the cell, after discharge at a specific current 150 mA g$^{-1}$ (**b**) and after discharge at a specific current 600 mA g$^{-1}$ (**e**). **c**, **f** 1D $^{23}$Na MR profile of metallic and dielectric sodium in the cell, after discharge at a specific current 150 mA g$^{-1}$ (**c**) and after discharge at a specific current 600 mA g$^{-1}$ (**f**). This cell was assembled with two separators.

Galvanostatic plating was continued, at a higher specific current of 600 mA g$^{-1}$, to promote further growth of dendrites. In the $^{23}$Na NMR spectra, 2D images and 1D profiles (Fig. 3) for the cell, following this high discharge rate, it can be seen that the signal for metallic Na (Fig. 3d) at 1145 ppm is now the dominant signal. This increase in signal intensity is also observed by the carbon WE at $z = 0.0$ mm, in both the 2D image (Fig. 3e) and 1D profile (Fig. 3f), and can be seen to have spread into the separators. This significant increase in signal, relative to that of the sodium CE, is because of the increased surface area expected for these dendritic structures[27,31]. Therefore, unlike a bulk metal surface, the NMR signal from dendrites increases proportionally to the mass deposited[30]. Lastly, the $^{23}$Na spectrum for the solvated Na species (Fig. 3d) reveals the disappearance of the broad peak at 1 ppm, an increase in the intensity of peak at 0 ppm and the continued presence of the peak at $-7$ ppm.

As the spatial resolution of the $^{23}$Na MR images is not sufficient to enable direct visualisation of the microstructure of this new metallic sodium environment, its development is observed (Fig. 4), indirectly, using $^1$H MRI[12]. In Fig. 4b, c, selected image planes from the 3D dataset are presented, before and after galvanostatic plating. The orientation of these 2D planes are referenced using a 3D surface rendering of the $^1$H signal from the electrolyte within the separator stack. Before plating (Fig. 4b), both vertical and horizontal images show relatively uniform fluid distribution, with the exception of a few bubbles of argon gas trapped during saturation of the separators. After plating (Fig. 4c), the vertical image reveals the dark silhouettes of several dendrites that have grown from the bottom of the cell into the separator stack. Likewise, the horizontal slice shows a large number of dark spots, which are the cross-sections of dendrites that have grown into that region. These dendritic structures can be seen more clearly in the "negative images" from the $^1$H 3D MRI data, which

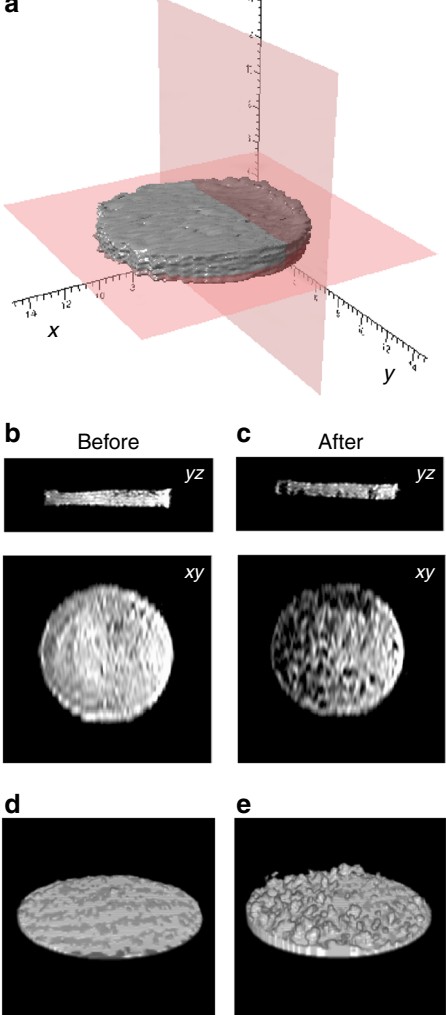

**Fig. 4 ¹H MRI of electrolyte signal before and after galvanostatic plating of the working electrode. a** 3D surface rendering from ¹H MRI of electrolyte in the separators before plating. The labelled red planes indicate selected 2D imaging slices from inside the cell. The vertical slice (*yz*) is at the centre of the separator stack and the horizontal slice (*xy*) is immediately above the carbon WE. Image slices are shown in columns **b** before and **c** after plating. Negative space images from 3D ¹H MRI of electrolyte **d** before and **e** after galvanostatic plating.

reveal the presence of metallic sodium, through the absence of the ¹H signal from the electrolyte. Figure 4 shows these negative images before (Fig. 4d) and after (Fig. 4e) plating, revealing how an initially smooth WE surface is later covered by a large number of dendrites of various sizes and shapes. These images reveal the coverage is not uniform, with a higher density of dendrite growth on the left side of the WE (as it is orientated in Fig. 4) and mostly smooth surface on the right.

In these metal cell experiments, ²³Na NMR spectroscopy has revealed the development of a range of new sodium environments during charge cycling and ²³Na/¹H MRI has revealed the formation and distribution of dendrites at high discharge rates. However, the formation of quasimetallic species in the carbon was not observed, this is most likely because of the low level of sodium transfer, and the second sodiation species normally observed at the low voltage plateau does not occur, as can be seen in Supplementary Fig. 2, where only a quarter of the capacity is observed. The use of metallic sodium in these cells also has the

disadvantage of potentially masking the formation of new metallic sodium environments, prior to dendrite formation, because of the signal of the metallic electrode dominates this region of the spectrum. Hence, to get around these problems, we removed the metallic sodium electrode and prepared full-cells, comprising a hard carbon anode and O3-type mixed metal oxide cathode. Here, we balanced the anode to cathode in the test cell with a cathode excess, and then cycled the cell with respect to Coulombic charging and discharging of the anode. This removes the resistances observed from the sodium metal anode in a two electrode configuration[32] and enables the full sodiation of the hard carbon, which is not possible in a cathode limited cell. Supplementary Figure 10 shows the electrochemical charge and discharge of the constructed full-cell and the expected respective anode and cathode voltages vs. Na/Na⁺.

**In operando monitoring in sodium full-cells**. The formation and development of sodium species in the full-cell was monitored in operando, during a charge/discharge cycle. A time series of ²³Na spectra were acquired during the formation cycle, which are shown as a 2D intensity plot in Fig. 5. Although these spectra were collected with a spectral width (2360 ppm), sufficiently large to capture both the metallic and solvated regions of the spectrum, these regions are presented separately (Fig. 5a, b). Selected, full spectra are shown in Fig. 6. The ²³Na NMR spectrum of the electrolyte, in the pristine full-cell (Fig. 6a), shows three distinct sodium environments, corresponding to the Na electrolyte in the separator (8 ppm), carbon in the electrodes (−7 ppm) and excess electrolyte around the electrode sandwich (0 ppm). These sodium environments evolve during charging, with additional sodium environments appearing <0 ppm, as previously observed by Stratford et al.[13]. The formation sodiation was also studied at lower specific current (30 mA g⁻¹) (Supplementary Fig. 3b). The sodiation of the hard carbon in the full-cell (Fig. 5c, Supplementary Fig. 10c) can be related to the sodiation voltages observed in the sodium metal cell (Supplementary Fig. 10a). Between 0–2.5 V cell voltage, the first stage of sodiation, which is equivalent to the sloping voltage region in the hard carbon half-cell (2.0–0.02 V vs. Na/Na⁺). Between 2.5 and 3.7 V cell voltage, the 2nd stage of hard carbon sodiation occurs, this is observed as the voltage plateau 0.2–0.01 V vs. Na/Na⁺. Figure 5 shows these two phases of sodiation: stage 1, evolution of the peaks around 0 ppm, up to 4 h into charging (~150 mA hg⁻¹ carbon), are indicative of sodium insertion and SEI formation, and is completed around a cell voltage of 2.6 V (Supplementary Fig. 10); stage 2, between 4 and 8 h up to 3.7 V (300 mA hg⁻¹ carbon) the formation of additional sodium environments are detected in the region where quasimetallic sodium nanoparticles (sodium pooling) have been previously observed[13,16]. This signal becomes apparent after 4 h and appears to shift and grow over time during charging. However, more surprisingly, at early stages of sodium insertion additional peaks appear in the metallic region of the ²³Na NMR spectrum (>1000 ppm). This was, therefore, studied further at a lower specific current of 30 mA g⁻¹ for the 1st charge or formation charge. Between 1.5 and 2.5 h, at a specific current 30 mA g⁻¹, the metal peak, is found to appear at 1050 ppm, and then shift to 1115 ppm, before disappearing below the level of detection. This metallic sodium formation during the first cycle has not been previously observed, but has been previously hypothesised[33]. It should be noted that these peaks would not have been detectable where a metallic sodium counter electrode was used, because their signal would have obscured by the large signal from the metallic electrode.

To investigate sodium plating, and the persistence of the sodium clusters, a new full-cell was charged to 4.18 V, where

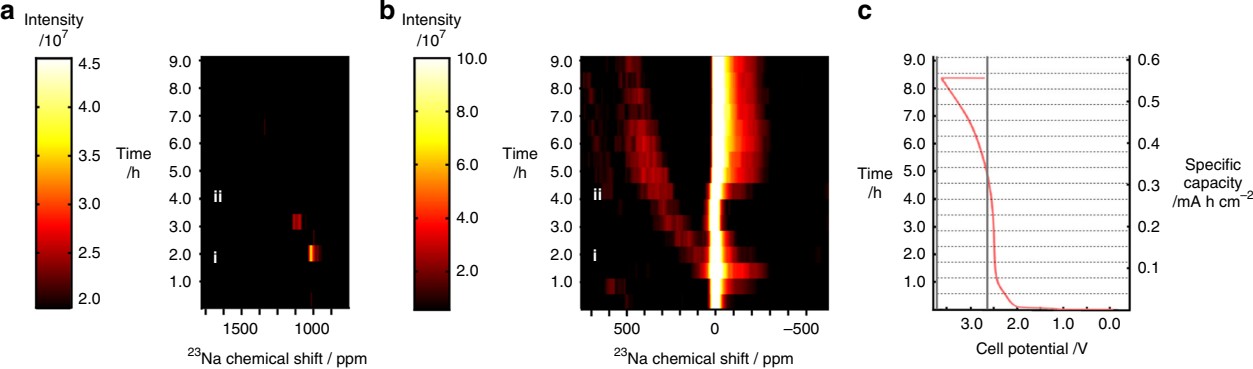

**Fig. 5 In operando $^{23}$Na NMR spectra and electrochemistry during a formation charge/discharge cycle.** 2D intensity plots of $^{23}$Na chemical shift, for regions around metallic signal (**a**) and solvated sodium (**b**), as a function of time, during the formation charge/discharge cycle at a specific current of 30 mA g$^{-1}$. **c** Corresponding charge profile of the formation cycle. Spectra for time points i and ii, indicated in **a** and **b**, are shown in Fig. 6b and c, respectively.

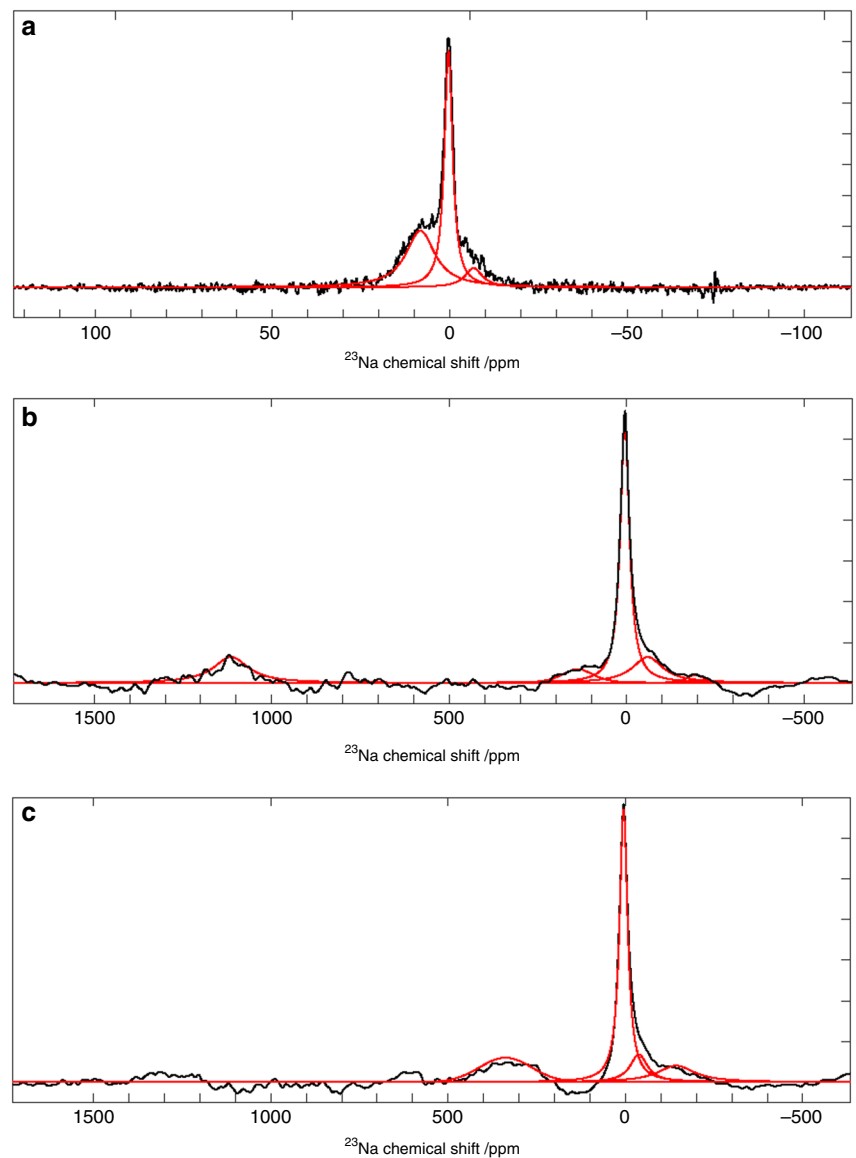

**Fig. 6 Selected in operando $^{23}$Na NMR spectra for a NIB full-cell during formation cycle. a** $^{23}$Na NMR spectrum from electrolytic region for pristine ($t = 0$) cell. Full $^{23}$Na NMR spectra 2.0 h (**b**) and 4 h (**c**) after charging, at a specific current of 30 mA g$^{-1}$, started. Experimental spectra are shown in black and the fitted peaks are given in red. Deconvolution values for these spectra can be found in Supplementary Table 6.

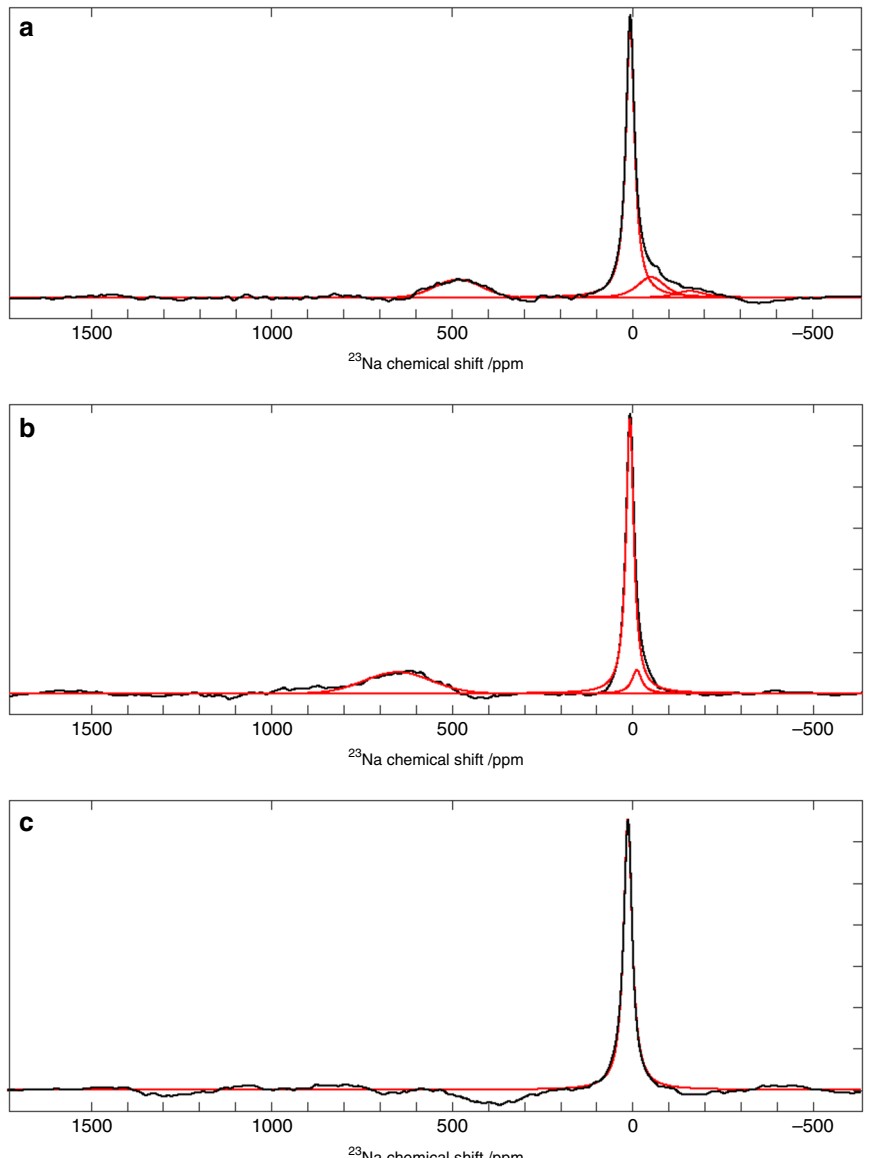

**Fig. 7 In situ $^{23}$Na NMR spectra for a NIB full-cell charged to 4.18 V at 86 mA g$^{-1}$.** $^{23}$Na NMR spectra immediately after charging (**a**), 4 h after charging (**b**) and after discharging (**c**). Experimental spectra are shown in black and the fitted peaks are given in red. Deconvolution values for these spectra can be found in Supplementary Table 5.

plating is believed to occur (Supplementary Fig. 10). In situ $^{23}$Na NMR spectra were acquired, which are shown in Fig. 7, immediately after the cell was charged, 4 h after charging and after the cell was discharged. The spectra after charging (Fig. 7a, b) reveal signal in a region of the spectrum (>400 ppm) where quasimetallic clusters were observed in operando, however, no metallic sodium signal is observed. This absence of plating is possibly caused by irreversible capacity within the cell, which utilises the excess sodium during the SEI formation on the hard carbon. These spectra also reveal an evolution of these sodium species, shown by the increase in chemical shift of this peak after 4 h, suggesting an increase in the size of the clusters[16]. This quasimetallic peak disappears after the battery is discharged (Fig. 5c). The $^{23}$Na NMR spectra also show new sodium environments in the solvated region of the spectrum following charging, as seen previously, where a distribution of peaks is observed down to −160 ppm. These environments are also seen to evolve over time, where after 4 h these peaks mostly disappeared.

## Discussion

We have performed, for the first time, operando $^{23}$Na NMR spectroscopy and $^{1}$H and $^{23}$Na MRI during galvanostatic cycling and plating in both full-cells and sodium metal cells. The resulting images and spectra reveal chemical changes for both metallic and solvated Na, which are linked to changes in the distribution of Na species within the cell. We have visualised the formation of sodium dendrites using both direct ($^{23}$Na) and indirect ($^{1}$H) MRI methods. $^{23}$Na NMR spectroscopy has revealed the development of both metallic and quasimetallic sodium, during charge cycling, as well as new solvated sodium environments during charge cycling, which have been associated with intercalated sodium, within the non-graphitic carbon working electrode, and a less mobile phase possibly associated with the SEI. It is important to note that the metallic signal, observed at the early stages of cycling, would not be visible in a sodium metal cell, as the enormous signal from the sodium electrode will have swamped these smaller signals. Only by going to the full-cell has this observation been possible. These new

metallic, quasimetallic and solvated Na environments require further work to fully establish their composition and location. Regardless, this study has revealed unique and new insight into the electrochemical behaviour of a sodium-ion battery, demonstrating the potential for such techniques to extract essential information required to better understand the mechanisms controlling performance and degradation in NIBs. Moreover, these experiments have been performed using off-the-shelf MRI probes, rather than specialist (purpose-built) equipment, demonstrating the potentially widespread accessibility of this technique. We, therefore, believe that a combination of operando $^1$H and $^{23}$Na NMR spectroscopy and imaging is able to reveal new insight in future studies, which is critical for establishing new battery materials and electrochemistries leading to the development of improved, next-generation batteries.

## Methods

**Electrode, electrolyte and cell assembly**. Sodium metal cells were constructed using a Swagelok cell design (Supplementary Fig. 1b). A sodium metal disk (1 mm thick, sodium ingot, 99.8% metals basis, Alfa Aesar) was used as the counter electrode (CE) and a disordered, non-graphitic carbon (hard carbon) anode material (1–2 mg cm$^{-2}$) was used for the working electrode (WE), which were separated by a number of electrolyte-saturated Whatman GF/B glass microfibre separators (675 μm); initially five (Fig. 1) in an uncycled cell, then two (Figs. 2–4) to study the dendrite growth to prevent short circuiting of the cell. Aluminium current collectors were used at both electrodes. The active carbon material in the anode was made by hydrothermal carbonisation at 230 °C for 12 h of a 10 wt% glucose (D-(+)-glucose, ≥99.5%, Sigma Aldrich) solution with the pH adjusted to 1–2 by the addition of citric acid (≥99.5%, Sigma Aldrich)[34], followed by high temperature annealing at 1300 °C under nitrogen for 2 h to increase conductivity. Electrodes were prepared from slurries (85 wt% hard carbon, 10 wt% sodium carboxymethyl cellulose binder ($M_w$~250,000, Sigma Aldrich) and 5 wt% carbon black (Super P conductive, 99+%, Alfa Aesar) in water) applied in a 250 μm layer on Al foil (conductive carbon coated, 18 μm, MTI corporation) (final active mass of 2.72 mg). All electrodes and separators had a diameter of 9.5 mm. The electrolyte and sodium metal cell preparation were conducted in an argon-filled glove box (Mbraun, H$_2$O < 0.5 ppm, O$_2$ < 0.5 ppm). Sodium hexafluorophosphate (NaPF$_6$, Alfa Aesar, 99+%) was first dried at 80 °C under vacuum for 16 h. A 1:1 mixture, by volume, of ethylene carbonate (EC, anhydrous 99%, Sigma Aldrich) and dimethyl carbonate (DMC, 99.9+%, Sigma Aldrich) was dried over 4 Å molecular sieves (20 vol%) for one day before use, and the solvent extracted from the top fraction to avoid contamination. A 1 M electrolyte solution was prepared by stirring NaPF$_6$ in EC/DMC until fully dissolved. Sodium-ion full-cells were constructed using a Swagelok cell design (Supplementary Fig. 1c), with a NaNi$_{1/3}$Fe$_{1/3}$Mn$_{1/12}$Sn$_{1/12}$O$_2$ (NaNFMSO) cathode[35,36] (4.5 mg cm$^{-2}$) hard carbon anode (1.5 mg cm$^{-2}$) separated by a single electrolyte-saturated Whatman GF/B glass microfibre separator (675 μm thick). Electrodes and separator had a diameter of 9.5 mm and aluminium current collectors were used at both electrodes. 100 g of the O3-type layered oxide was prepared by solid-state reaction the details of which are published elsewhere[36]. The fired sample was transferred to a glove box (MBraun, H$_2$O < 0.1 ppm, O$_2$ < 0.1 ppm) milled and sieved before making into an electrode. A positive electrode slurry of 87% Active material (NaNFMSO): 6% Carbon (C65, Imerys): 7% Binder PVDF (Polyvinylidene fluoride) (Solef$^\circledR$ 5130, battery grade) in NMP (N-methyl-2-pyrrolidone) (Sigma Aldrich, 99.5%) was coated onto aluminium current collector using a doctor blade technique. The hard carbon anode was prepared using a similar slurry method to the cathode. The slurry containing 90 wt% hard carbon, 5 wt% conductive carbon, and 5 wt% binder was coated on Al foil. Both electrodes were dried on a hotplate at 80 °C with final drying carried out in a vacuum oven at 120 °C overnight. Cells were assembled as shown in Supplementary Fig. 1c, in a dry room (dew point of −50 °C) and the cell was filled with an electrolyte containing 1 M NaPF$_6$ in EC and DEC (diethyl carbonate) (1:1 by volume) (Fluorochem).

**In situ magnetic resonance measurements**. $^{23}$Na NMR spectroscopy and $^{23}$Na and $^1$H MRI were performed on a Bruker 9.4 T spectrometer equipped with either a $^{23}$Na, $^1$H or $^1$H/$^{23}$Na 25 mm WB40 radiofrequency (RF) probes. Details of sample positioning and calibration of parameters can be found in Supplementary Methods. For the sodium metal cell experiments, $^{23}$Na NMR spectra were acquired separately at two resonance frequencies: one for Na$^+$ ions in the electrolyte (0 ppm) and the other for the Na metal (1131 ppm). A repetition time of 0.1 s and 2048 averages were used, resulting in an experiment time of 3 min 24 s. Two-dimensional (2D) $^{23}$Na MR images of the electrolyte and metal were acquired, separately, using a spin-echo sequence at different reference frequencies (see Supplementary Methods). Owing to short NMR relaxation times, no slice selection was used, hence images were projections of the entire cell thickness onto a 2D plane. All images have frequency encoding in the vertical direction ($z$, along the

axis of the cell) and phase encoding in the horizontal direction ($x$), with a pixel size of 231 μm ($z$) × 1050 μm ($x$). A repetition time of $T_R$ = 0.1 s was used with 1024 averages, which required 28 min for each image. One-dimensional (1D) profiles were acquired along the vertical dimension only. These were collected using 2048 signal averages and a repetition time of $T_R$ = 0.1 s, resulting in an experiment time of 3 min 35 s. Three-dimensional (3D) $^1$H MR images of the electrolyte in the sodium metal cell were acquired using a multi-slice spin-echo sequence with horizontal frequency encoding, 15 mm isotropic field-of-view, with a 117 μm × 117 μm × 469 μm voxel size, and a 1.0 s repetition time. Zero-filling in the third dimension was used during processing to produce 3D images with isotropic voxels. "Negative images" of sodium microstructures were produced from the $^1$H 3D MRI data by only displaying voxels below a threshold intensity (see Supplementary Methods), which identified regions where the electrolyte was absent, or at least significantly depleted and below the level of detection, and hence enabled the detection of metallic sodium microstructures within the electrolyte. For the full-cell measurements, $^{23}$Na NMR spectra were acquired with a sufficiently large spectral width to capture signal from sodium in solvated and metallic environments. A spin-echo sequence was used, with pulse lengths of 2 and 4 μs duration, a spectral width of 2360 ppm, collecting 512 points (zero-filled to 2048), with a repetition time of 0.05 s and 32768 averages, resulting in an experiment time of 27 min 18 s. In operando measurements were collected continuously during charging, with a delay of 300 s in between each spectrum.

**Electrochemical measurements**. Electrochemical measurements were performed using an Ivium Octostat 5000 potentiostat connected to the cell in a 2-electrode configuration. Cycling of cells inside the MRI magnet utilised cable connections, low-pass filtering and shielding, which are all described previously[37]. For the sodium metal cell, an open circuit voltage was recorded and an initial discharge/charge cycle at a specific current of 30 mA g$^{-1}$ was performed outside the magnet before imaging (Supplementary Fig. 2). A second charge cycle (Supplementary Fig. 2) was performed inside the magnet at a specific current of 20 mA g$^{-1}$, to optimise the cycling time for the imaging experiment. Following this, the effect of galvanostatic plating on the carbon WE was investigated by discharging the cell at 150 mA g$^{-1}$ for a period of 137 min ($E$ < 0.001 V, 0.572 mA cm$^{-2}$, total charge 0.931 mA h). Discharging was then continued at a higher specific current (maximum 600 mA g$^{-1}$; 1.632 mA cm$^{-2}$), to promote further growth of the dendrites, for a period 210 min (total charge 6.308 mA h). For the full-cell experiments, an open circuit voltage was recorded and a formation cycle (Supplementary Fig. 3) was performed inside the magnet (in operando) at a specific current of 30 mA g$^{-1}$, while $^{23}$Na NMR spectra were acquired every 33 min, as well as a charge/discharge cycle outside the magnet at a specific current of 86 mA g$^{-1}$, before in situ $^{23}$Na NMR spectroscopy.

## Data availability

The data generated in this study are available at https://doi.org/10.25500/edata.bham.00000456.

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

## Acknowledgements

We thank Peter Slater for helpful discussions. M.M.B., G.E.P. and T.M. thank the University of Birmingham and University of Nottingham for funding through the Strategic Collaborative Fund. C.L.D. thanks EPSRC (EP/N509590/1) for funding. M.-M.T. and H.Au thank EPSRC (EP/R021554/1) for funding. H.Al. thanks the Turkish government for a PhD scholarship. G.E.P. and T.M. thank the Medical Research Council for funding (Grant No. MC_PC_15074).

## Author contributions

M.M.B. and G.E.P. designed the research and performed NMR/MRI experiments. J.M.B. and C.L.D designed the swagelock cells and planned and performed in operando electrochemical and MRI experiments. M.T., H.Au and H.Al. designed and synthesised the carbon electrode material, for the sodium metal cell experiments, and carried out its physical and electrochemical characterisation. E.K., L.C. and B.K. designed, synthesised and characterised the NaNFMSO and hard carbon electrodes for the full-cell experiments. G.E.P. and T.M. developed the $^{23}$Na NMR and MRI protocols. C.L.D. performed $^{23}$Na NMR deconvolution. J.M.B. and M.M.B. wrote the manuscript. M.M.B., G.E.P., E.K., M.T. and T.M. supervised the research. All authors discussed the results and commented on the manuscript.

## Competing interests

The authors declare no competing interests.
