## [Peer Review File · Nature Communications]

Reviewers' comments:

Reviewer #1 (Remarks to the Author):

This paper presents an advanced analysis and in operando imaging of a sodium cell based on NMR. The experiment is well designed to observe the response during cycling of a carbon electrode in absence and in presence of Na plating. I recommend publication of this article after the following questions will be addressed:

The coulombic efficiency obtained in the in-situ cell (fig. S12) is very low (~10% at 1st cycle). From the scheme in figure S11, it can be seen that there is an important dead volume on the working electrode side, that will be filled with electrolyte.

It can be expected that the stack pressure will be relatively low, and there might be some concern regarding the the air-tightness of the polymer cell body, which can usually not be closed as tightly as a metallic cell. For these reasons, a significant amount of irreversible side reactions can be expected in such a cell, as well as an important internal resistance. It is therefore likely that the state of charge of the carbon electrode is still very low as the cut-off voltage is reached because of the IR drop, and that the 10% reversible capacity correspond to the actual amount of Na inserted in the carbon at 1st charge. However, the authors state that "a considerable amount of sodium remains trapped in the carbon", on which basis is it discarded that it would be a considerable amount of irreversible loss to side reactions, such as electrolyte decomposition?

How come cycle 2 starts at 0.5V when the sample was oxidized up to 2.5V at the end of 1st cycle?

In fig.3, the presence of metallic Na is observed on the working electrode "at the start" of galvanostatic plating. What does "at the start" exactly mean? Though a relatively high rate of C/2 is used, one might indeed be surprised that metallic sodium is present on the working electrode as soon as the current is applied. It is usually possible to detect the nucleation peak on the voltage curve when plating occurs, was it the case here? Could this voltage curve corresponding to figure 3 be added in the supplementary information?

The peak at 0 ppm is attributed to solvated Na species present in the separator between the two electrodes, whereas the peak at -4 ppm is attributed to excess electrolyte outside of the electrodes and stack. Neither of these two contributions would be expected to significantly change during cycling since the Na counter electrode will compensate for any Na consumption from the electrolyte. In this context, what is the reason for the peak at 0 ppm to significantly decrease at the beginning of galvanostatic plating (figs 3a and S15a) and then increase again for continued plating (in figs 3a and S15b)?

A detail should be corrected in the list of references, where ref. 10 is duplicate of ref. 5.

Reviewer #2 (Remarks to the Author):

This is an interesting manuscript by Britton and co-authors, however, in my opinion, it does not meet the criteria for novelty required for Nature Communications. The main problem of the method is associated with the fact that it requires a special cell design, which varies substantially from a conventional battery. It significantly limits the practical application of obtained results; and in my opinion, this should be a focal point of further method development, such that results coming forward actually contribute to the understanding of relevant electrochemical cells.

The current manuscript does not provide any benefits in that sense:

- 1) The authors utilize a traditional cell design with electrode separation of 2.5-5 mm which is two orders of magnitude larger than that of commercial batteries;
- 2) The cell's electrochemical behavior is not even close to traditional batteries with 90% irreversible capacity loss at $c/10$ cycling and even more drastic changes at $c/2$. It is hard to conclude that any meaningful electrochemical data can be extracted from such a cell.
- 3) There is nothing new in the MRI techniques as presented, moreover the frequency encoding pulse sequence is the worst choice for such experiments, because it is highly sensitive to magnetic field inhomogeneity caused by susceptibility variation in the cell, leading to signal attenuation in problematic regions up to a factor 20 (Proc Nat Acad Sci 113, 10779-10784);
- 5) The authors are pointing out that ^{23}Na nucleus has faster relaxation than ^7Li , and declare that ^{23}Na is more technically demanding because of that, however the shortest T_2^* reported in supplementary information is 0.38 ms, and it was shown previously that high resolution operando images can be obtained for lithiation of graphite electrodes with 4 times shorter ^7Li T_2^* (0.1 ms; J. Phys. Chem. C 2018, 122, 21784–21791).

While these were general comments regarding the topic, I also have many specific questions and suggestions:

- 1) Lines 109-110. It is said that either 5 (figure 1) or two (figures 2-5) separators (280 μm thick) were placed between electrodes. However, the distance between electrodes on Fig. 1 is about 3.7mm, which is 2.5 times larger than $5 \times 0.28 = 1.4$ mm. Even larger discrepancies arise for Fig. 2-5, where the distance between electrodes is about 2.2 mm instead of suggested 0.56 mm.
- 2) Lines 138-141. The fact that the MR signal in a voxel is lower than a threshold does not guarantee that the electrolyte was absent in the corresponding region, such attenuation can be caused by B_0 distortion due to an appearance of Na dendrite there (it should not fill the whole voxel: Proc Nat Acad Sci 113, 10779-10784).
- 3) Lines 150-153. It is written that the cell was discharged at $C/2$ rate (0.572 mA/cm^2) for 137 minutes. It means that the carbon was sodiated for $137/120 \times 100\% = 114\%$. The total charge is stated as 0.931 mAh (therefore $C = 0.815$ mAh), however according to the Fig. S12 the cell capacity is 0.11 mAh. Could you explain the disagreement? Then, it is written, that discharging was continued at $2C$ rate (1.632 mA/cm^2). How is it possible that C -rate was changed 4 times (from $C/2$ to $2C$), but current density was changed only 2.85 times (from 0.572 to 1.632)?
- 4) Lines 184-186. It is pointed out that there is a blurring of collected images. Particularly, looking to fig. 1d one can see that edges of electrolyte domain have a slope on both sides propagated for about a 1 mm each along z axes. Does this mean that this is an actual resolution in this dimension instead of nominal 0.231 mm calculated as FOV/TD ?

Overall, this is an interesting study, however due to extensive previous work in this area, my evaluation is that the manuscript does not meet the criteria for novelty and impact required by Nature Communications.

Reviewer #3 (Remarks to the Author):

This article reports an MRI research of sodium ion batteries consisting of non-graphitic carbon electrode and Na metal electrode. Authors performed ^{23}Na and ^1H NMR and MRI measurements using an originally assembled Swagelok cell. In situ analyses are essentially important to observe non-equilibrium state during charge and discharge of batteries, and this research significantly contributes the development of in MRI technique. A distinguishing of sodium in electrolyte in and outside electrode (including in separator) is useful. However, the reviewer feels some issues which should be corrected or be discussed before publication, for example, observation of Na clusters by NMR, and benefit of MRI method.

Comments to the articles are as follows:

1. In the title and the manuscript, authors repeatedly use a word "operando", but it is not correct. "Operando" means 'in operation', namely, the word means "measurement during working of a battery (during charge or discharge)". However, authors only show NMR spectra and MR images of static states before and after charging of the battery. In such a case, "in situ" might be acceptable for readers, but "operando" is not acceptable.

2. A part of explanation about previous researches in the Introduction is inexact. In page 3, authors write "Operand solid-state NMR studies of hard carbons(17) ...", but ref. 17 doesn't use operando analysis. The reference should be replaced to ref. 14.

Authors also write "Comparison of these spectra with ex situ MAS NMR ...found to disappear over a few hours(17), which explains... in previous work." in page 3. However, ref. 17 doesn't report about disappearing of signals directly. Recently the authors of ref. 17 reported that a usage of FEC additive prevents a formation of quasimetallic Na clusters (shift to higher than 600 ppm). (*)

The reviewer agrees that in situ experiments of battery samples are effective to measure fresh samples, but the incorrect description in the introduction should be corrected.

(*) Carbon, 145, 712-715 (2019).

3. There is no data about "non-graphitic carbon anode material" although the precursor and the carbonization condition is described in the manuscript. Information of the sample such as X-ray diffraction pattern and specific surface area is in demand because the electrochemical capacities and properties of anode strongly depend on the structure of carbon materials.

4. About charged sample (in Figure 2), authors illustrate NMR spectra around 1130 ppm and 0 ppm, but there is no signal about quasimetallic Na clusters which has been reported in ref. 14 and above reference (*). Why isn't it observed?

5. Author mentions that "The presence of similar additional peaks has not been observed previously during galvanostatic cycling in the in situ ^{23}Na NMR study ...". However, at least, signal of irreversible sodium ascribable to SEI has been reported in an ex situ ^{23}Na NMR study. (**)

(**) J. Mater. Chem. A., 4, 13183-13193 (2016).

6. 3D surface rendering from ^1H MRI is a unique and very interesting attempt to observe the silhouette of Na metal dendrite. Interestingly, the dendrite structure seems to be very bulky (in Fig. 4e). Have authors observed the dendrite structure on the electrode directly, after plating of the electrode and disassembling the cell?

7. The reviewer feels that ^{23}Na MRI technique is useful to understand the state of sodium in SIB.

However, advantages of ^{23}Na and ^1H MRI in comparison with the other in situ (operando) analyses is not clear. For example, the resolution of the authors' experiments seem to be still lower than that of in situ electron microscope technics (in situ TEM, in situ SEM and their EDX analysis), and seem to be difficult to exceed the resolution of microscope in future. In situ analyses using X-ray (XAFS and XRD) are also very developed recently.

What will be able to become a strong merit of ^{23}Na MRI and ^1H MRI in future? The issue should be discussed in comparison with the other in situ analyses in the manuscript, for readers to understand the significance of the authors' research.

Reviewers' comments:

Reviewer #1 (Remarks to the Author):

This paper presents an advanced analysis and in operando imaging of a sodium cell based on NMR. The experiment is well designed to observe the response during cycling of a carbon electrode in absence and in presence of Na plating. I recommend publication of this article after the following questions will be addressed:

The coulombic efficiency obtained in the in-situ cell (fig. S12) is very low (~10% at 1st cycle). From the scheme in figure S11, it can be seen that there is an important dead volume on the working electrode side, that will be filled with electrolyte.

It can be expected that the stack pressure will be relatively low, and there might be some concern regarding the the air-tightness of the polymer cell body, which can usually not be closed as tightly as a metallic cell. For these reasons, a significant amount of irreversible side reactions can be expected in such a cell, as well as an important internal resistance. It is therefore likely that the state of charge of the carbon electrode is still very low as the cut-off voltage is reached because of the IR drop, and that the 10% reversible capacity correspond to the actual amount of Na inserted in the carbon at 1st charge. However, the authors state that "a considerable amount of sodium remains trapped in the carbon", on which basis is it discarded that it would be a considerable amount of irreversible loss to side reactions, such as electrolyte decomposition?

How come cycle 2 starts at 0.5V when the sample was oxidized up to 2.5V at the end of 1st cycle?

Authors response: The reviewer makes several good points about the electrochemical behaviour of the half-cell. There do indeed seem to be a number of issues with the performance of this battery, which we believe are caused by high internal cell resistances, as also suggested by the reviewer. For these reasons, we moved away from this half-cell configuration to a full-cell which has improved the electrochemical performance of the cells we have been studying. We believe that in this sodium metal anode cell we are observing sodium metal plating rather than observing the mechanism of sodium intercalation into the hard carbon. We have clarified these points in the text.

In fig.3, the presence of metallic Na is observed on the working electrode "at the start" of galvanostatic plating. What does "at the start" exactly mean? Though a relatively high rate of C/2 is used, one might indeed be surprised that metallic sodium is present on the working electrode as soon as the current is applied. It is usually possible to detect the nucleation peak on the voltage curve when plating occurs, was it the case here? Could this voltage curve corresponding to figure 3 be added in the supplementary information?

Authors response: We have improved the electrochemical test vehicles and therefore the text has been updated accordingly. The metal anode cell we see a short period of sloping voltage profile, which has been previously discussed as intercalation between layers, and then due to the high polarisation we observed sodium plating as the voltage reaches close to 0 vs Na/Na+.

Interestingly in the full cell arrangement, we do see a sodium metallic peak upon initial sodiation of the hard carbon, this is indeed very surprising. This has only been observed in

the full cell arrangement as it is possibly masked by the sodium in a metal anode cell. This has been discussed as potentially possible previously by Palacin, Tarascon et al (2016) and was observed as a current spike at start of charge. We do not however see sodium 'plating' as such in the full cell, because the balance of the anode to cathode is 1:1.3 and therefore the level of sodium plating is relatively small.

The peak at 0 ppm is attributed to solvated Na species present in the separator between the two electrodes, whereas the peak at -4 ppm is attributed to excess electrolyte outside of the electrodes and stack. Neither of these two contributions would be expected to significantly change during cycling since the Na counter electrode will compensate for any Na consumption from the electrolyte. In this context, what is the reason for the peak at 0 ppm to significantly decrease at the beginning of galvanostatic plating (figs 3a and SI5a) and then increase again for continued plating (in figs 3a and SI5b)?

Authors response: We were puzzled by this, but can only suggest that it is possible there is a change in the nature of the solvated species (leading to a change in the spectrum) or the rate at which the sodium is plated may be initially too high for the sodium in the electrolyte to be replaced, but as plating continues, the amount of sodium in the electrolyte recovers.

A detail should be corrected in the list of references, where ref. 10 is duplicate of ref. 5.

Authors response: we have corrected this now.

Reviewer #2 (Remarks to the Author):

This is an interesting manuscript by Britton and co-authors, however, in my opinion, it does not meet the criteria for novelty required for Nature Communications. The main problem of the method is associated with the fact that it requires a special cell design, which varies substantially from a conventional battery. It significantly limits the practical application of obtained results; and in my opinion, this should be a focal point of further method development, such that results coming forward actually contribute to the understanding of relevant electrochemical cells.

Authors response: While we have used a 'special cell design', this is a requirement for other operando measurements such as SANS, EPR, Raman, XAS and TEM/SEM. However, our electrode sandwich does resemble closely what is found in a coin cell. However, we have done as the reviewer suggested and undertaken further method development and extended our study to include investigation of a full-cell. While the orientation of current collector, electrode and separator in our Swagelok design does not differ from a coin cell, we have further optimised our cell design to bring it closer to that more typically utilised by battery researchers. In our latest (full-cell) version, we have reduced the number of separators and removed the springs, to ensure compression is more uniform across the cell. We have also engineered the cell to minimise resistances, and maximise what we observe on the hard carbon anode. The cell is now negatively limited rather than positively limited, and therefore we can study the imaging and spectra of the hard carbon sodiation mechanism with significantly greater control. The results achieved in this new cell has provided novel findings and new insight into the behaviour of sodium ion batteries.

The current manuscript does not provide any benefits in that sense:

1) The authors utilize a traditional cell design with electrode separation of 2.5-5 mm which is two orders of magnitude larger than that of commercial batteries;

Authors response: While we show a cell containing 5 separators, we do not perform electrochemical measurements on this system. The half-cell experiments employed two separators, with an electrode separation of 2.5 mm. However, our later results use a full-cell which only employs a single separator, which was compressed, leading to an electrode separation $< 1\text{mm}$, which is now closer to that used in traditional cells.

2) The cell's electrochemical behavior is not even close to traditional batteries with 90% irreversible capacity loss at $c/10$ cycling and even more drastic changes at $c/2$. It is hard to conclude that any meaningful electrochemical data can be extracted from such a cell.

Authors response: We have shown in this work that MRI techniques are possible for monitoring sodium ion cells. The initial work on the sodium metal anode cell, shows that sodium plating can be observed. The cell set-up was highly resistive and therefore the lower cut-off voltage reached before the full sodiation was achieved. However, with the new cell set-up we have minimised the resistances, and we have greater voltage control – in the supplementary data the full cell electrochemistry curve, is overlaid onto the hard carbon and layered oxide voltage curves vs Na/Na^+ . From the normalisation of the capacity (mAh/cm^2) the precise voltage for different aspects of the sodiation can be targeted. We have here shown 'in operando' several different key aspects of the sodiation mechanism using this cell.

3) There is nothing new in the MRI techniques as presented, moreover the frequency encoding pulse sequence is the worst choice for such experiments, because it is highly sensitive to magnetic field inhomogeneity caused by susceptibility variation in the cell, leading to signal attenuation in problematic regions up to a factor 20 (Proc Nat Acad Sci 113, 10779-10784);

Authors response: We disagree with the reviewer's statement that "There is nothing new in the MRI techniques as presented". While we are using well established MRI sequences, the application of ^{23}Na MRI for the 2D localisation of sodium, in operando, has not been previously reported and provides therefore a new pathway to study batteries. The additional results on the full-cell, which we present in this resubmission, provides both significant and new insight into the sodiation mechanism in hard carbons. Concerning the use of frequency encoding, we have found that where metallic electrodes are carefully aligned to be parallel with the B_0 field, negligible effects from magnetic susceptibility differences are observed. The advantage with using frequency encoding for the 1D profiles and a combination of frequency and phase encoding for the 2D images is that it is possible to acquire images more rapidly than typically possible by pure-phase encoding methods. Making it possible to observe changes during charge cycling in real-time.

5) The authors are pointing out that ^{23}Na nucleus has faster relaxation than ^7Li , and declare that ^{23}Na is more technically demanding because of that, however the shortest T_2^* reported in supplementary information is 0.38 ms, and it was shown previously that high resolution operando images can be obtained for lithiation of graphite electrodes with 4 times shorter ^7Li T_2^* (0.1 ms; J. Phys. Chem. C 2018, 122, 21784–21791).

Authors response: The reviewer is of course correct concerning the T2 relaxation times that appears to be more favourable for ^{23}Na MRI compared to that found for lithium in the previous work. However, that only strengthens the case presented here, because our materials demonstrated $T_2^* > 0.38\text{ms}$, this allowed us to apply this 2D visualisation methodology without any complications over a range of length scales that are a few orders of magnitude larger than those reported in J. Phys. Chem. C 2018, 122, 21784–21791. Because of this, the resolution in the images reported in the J. Phys. Chem C paper, which were limited to 1D profiles, are also higher than the resolution of ^{23}Na images that we report, but we acquired 2D images and imaged across a greater field-of-view. Moreover, we were able to accomplish, in operando, imaging using standard 25mm microimaging ^{23}Na probes that are much simpler and cheaper than the diff50 probe used in J. Phys. Chem. C 2018, 122, 21784–21791. That been said we have modified the text to make it more focussed on the context of the present 2D microimaging methodology. In fact, we took advantage of ^{23}Na faster T1 relaxation and were able to collect 2D sodium images in 28 min vs 1D 7Li in 2h (J. Phys. Chem. C 2018, 122, 21784–21791) thus making ^{23}Na methodology to be more suitable for finer time resolution needed to capture specific features of in operando performance of sodium batteries during charge/discharge cycles with the opportunity to localise both metallic and electrolytic sodium.

While these were general comments regarding the topic, I also have many specific questions and suggestions:

1) Lines 109-110. It is said that either 5 (figure 1) or two (figures 2-5) separators (280 μm thick) were placed between electrodes. However, the distance between electrodes on Fig. 1 is about 3.7mm, which is 2.5 times larger than $5 \times 0.28 = 1.4\text{ mm}$. Even larger discrepancies arise for Fig. 2-5, where the distance between electrodes is about 2.2 mm instead of suggested 0.56 mm.

Authors response: This has been clarified, and a new test method / cell set-up illustrated in the text

2) Lines 138-141. The fact that the MR signal in a voxel is lower than a threshold does not guarantee that the electrolyte was absent in the corresponding region, such attenuation can be caused by B0 distortion due to an appearance of Na dendrite there (it should not fill the whole voxel: Proc Nat Acad Sci 113, 10779-10784).

Authors response: While B0 distortions, and partial filling of voxels, may blur the edges of the dendrites, this will not have a significant impact on the visualisation of dendrites following plating in the sodium metal cell and are within the resolution needed for this work. Moreover, these 3D ^1H images compare well with the 2D ^{23}Na MR images.

3) Lines 150-153. It is written that the cell was discharged at C/2 rate (0.572 mA/cm²) for 137 minutes. It means that the carbon was sodiated for $137/120 \times 100\% = 114\%$. The total charge is stated as 0.931 mAh (therefore C = 0.815 mAh), however according to the Fig. S12 the cell capacity is 0.11 mAh. Could you explain the disagreement? Then, it is written, that discharging was continued at 2C rate (1.632 mA/cm²). How is it possible that C-rate was changed 4 times (from C/2 to 2C), but current density was changed only 2.85 times (from 0.572 to 1.632)?

Authors response: We have modified the text to mA/g which is more appropriate for this test.

4) Lines 184-186. It is pointed out that there is a blurring of collected images. Particularly, looking to fig. 1d one can see that edges of electrolyte domain have a slope on both sides propagated for about a 1 mm each along z axes. Does this mean that this is an actual resolution in this dimension instead of nominal 0.231 mm calculated as FOV/TD?

Authors response: The blurring seen in fig1d, along the z direction, is most likely to be caused by slight unevenness in the surface and that the electrode is not perfectly aligned along the z direction.

Overall, this is an interesting study, however due to extensive previous work in this area, my evaluation is that the manuscript does not meet the criteria for novelty and impact required by Nature Communications.

Authors response: While, we do not agree with the reviewer's comment about the novelty of the previous manuscript, the additional study on the full cell provides additional novelty. Firstly, we have presented results for a full cell (where previous NMR studies on NIBs have exclusively investigated sodium metal cells), as well as observing the formation of metallic sodium during the formation cycle, which has never been observed before.

Reviewer #3 (Remarks to the Author):

This article report an MRI research of sodium ion batteries consisting of non-graphitic carbon electrode and Na metal electrode. Authors performed ^{23}Na and ^1H NMR and MRI measurements using an originally assembled Swagelok cell. In situ analyses are essentially important to observe non-equilibrium state during charge and discharge of batteries, and this research significantly contributes the development of in MRI technique. A distinguishing of sodium in electrolyte in and outside electrode (including in separator) is useful. However, the reviewer feels some issues which should be corrected or be discussed before publication, for example, observation of Na clusters by NMR, and benefit of MRI method. Comments to the articles are as follows:

1. In the title and the manuscript, authors repeatedly use a word "operando", but it is not correct. "Operando" means 'in operation', namely, the word means "measurement during working of a battery (during charge or discharge)". However, authors only show NMR spectra and MR images of static states before and after charging of the battery. In such a case, "in situ" might be acceptable for readers, but "operando" is not acceptable.

Authors response: we show a range of experiments which include both in situ and in operando measurements. With the additional in operando measurements we have included on the full-cell, we believe we are justified in keeping "operando" in the title.

2. A part of explanation about previous researches in the Introduction is inexact. In page 3, authors write "Operand solid-state NMR studies of hard carbons(17) ...", but ref. 17 doesn't use operando analysis. The reference should be replaced to ref. 14.

Authors response: we have corrected this now.

Authors also write "Comparison of these spectra with ex situ MAS NMR ...found to disappear over a few hours(17), which explains... in previous work." in page 3. However, ref. 17 doesn't report about disappearing of signals directly. Recently the authors of ref. 17 reported that a usage of FEC additive prevents a formation of quasimetallic Na clusters (shift to higher than 600 ppm). (*)

The reviewer agrees that in situ experiments of battery samples are effective to measure fresh samples, but the incorrect description in the introduction should be corrected.

(*) Carbon, 145, 712-715 (2019).

Authors response: we have corrected this now.

3. There is no data about "non-graphitic carbon anode material" although the precursor and the carbonization condition is described in the manuscript. Information of the sample such as X-ray diffraction pattern and specific surface area is in demand because the electrochemical capacities and properties of anode strongly depend on the structure of carbon materials.

Authors response: These details have been included in the Supplementary details.

4. About charged sample (in Figure 2), authors illustrate NMR spectra around 1130 ppm and 0 ppm, but there is no signal about quasimetallic Na clusters which has been reported in ref. 14 and above reference (*). Why isn't it observed?

Authors response: We believe our half-cell wasn't sufficiently well optimised to give us the required electrochemical behaviour to observe these quasimetallic species. The internal cell resistances meant that the lower voltage cut-off was reached before full sodiation of the hard carbon was observed. Moving to the full-cell we have minimised the internal cell resistances, and been able to both observe quasimetallic species and follow their evolution. Additionally, by removing the sodium metal counter electrode, we have been able to observe the formation of metallic sodium during the sodiation of hard carbon working electrode. The search for these species, and the change in cell, forms a new additional section to the paper.

5. Author mentions that "The presence of similar additional peaks has not been observed previously during galvanostatic cycling in the in situ ^{23}Na NMR study ...". However, at least, signal of irreversible sodium ascribable to SEI has been reported in an ex situ ^{23}Na NMR study. (**)

(**) J. Mater. Chem. A., 4, 13183-13193 (2016).

Authors response: we have amended the manuscript accordingly.

6. 3D surface rendering from ^1H MRI is a unique and very interesting attempt to observe the silhouette of Na metal dendrite. Interestingly, the dendrite structure seems to be very bulky (in Fig. 4e). Have authors observed the dendrite structure on the electrode directly, after plating of the electrode and disassembling the cell?

Authors response: This is a great idea and something we thought about doing, but we were unable to disassemble the cell to observe the dendrites visually.

7. The reviewer feels that ^{23}Na MRI technique is useful to understand the state of sodium in SIB. However, advantages of ^{23}Na and ^1H MRI in comparison with the other in situ (operando) analyses is not clear. For example, the resolution of the authors' experiments seem to be still lower than that of in situ electron microscope technics (in situ TEM, in situ SEM and their EDX analysis), and seem to be difficult to exceed the resolution of microscope in future. In situ analyses using X-ray (XAFS and XRD) are also very developed recently. What will be able to become a strong merit of ^{23}Na MRI and ^1H MRI in future? The issue

should be discussed in comparison with the other in situ analyses in the manuscript, for readers to understand the significance of the authors' research.

Authors response: This a good point and we have added these details to the manuscript.

Reviewers' comments:

Reviewer #1 (Remarks to the Author):

The work reported in this manuscript was significantly improved, with new experiments carried out using better electrodes and a full cell configuration. Concerning the analysis of voltage profiles, a few points would still deserve some revision before publication.

The discussion of figures 2 and S12 remains misleading and partially erroneous. Indeed fig. S12, especially cycle 1, does not really look like the typical slope-plateau voltage profile for hard C, which would rather look like fig S110a in an ideal case, and like fig. S19 in an unoptimized cell. From fig. S12 it can be seen that the first discharge is stopped after (possibly incomplete) SEI formation, and definitely before the low-voltage plateau is reached. There is no evidence that any sodium remains trapped inside the carbon structure since the ~ 45 mAh g⁻¹ irreversible capacity can entirely be attributed to this SEI formation. On the other hand, it is actually interesting that at cycle 1 the analysis is carried after SEI formation only. As indicated in the description of fig. S15, no major changes in the distribution of Na species are seen within the cell, and the reason is most probably that no significant Na insertion in the carbon occurred since 30 mAh g⁻¹ reversible capacity corresponds to the very beginning of the slopy region.

The full cell is charged to 4.18 V, where plating is believed to occur. No sodium plating is actually seen in this cell and the authors mentioned in their rebuttal letter that plating is relatively small because of the 30% excess capacity of the cathode. However, it is likely that this capacity excess is compensated by the irreversible capacity at the anode, which is hardly reproducible from one type of cell to another, especially when moving to operando cells. Figure S110c illustrates well the purpose of the authors by assembling a full cell with overcapacitive cathode to reach Na plating on the anode. It also shows that the irreversible capacity is higher in the full cell, with more irreversible capacity on the anode side than in the half cell, since the last inflexion at 0.14 mAh cm⁻² in the C half-cell is seen later in the full cell, around 0.25 mAh cm⁻². This feature is not attributable to the cathode which shows a monotonous voltage trend vs capacity in fig. S110b. More irreversible capacity implies that plating will occur later, at a stage that possibly was not reached in the full cell. If reduction was extended below 0V in the C half-cell, Na plating would occur after a nucleation peak that would be easily identified in the voltage profile of a full cell as well. In absence of such peak during the charge of the cell, it can be believed that plating did actually not occur. Nevertheless, relevant information is obtained from this work regarding the state of Na in hard C at low voltage.

Reviewer #2 (Remarks to the Author):

The authors have addressed the concerns I had with the original version of the manuscript, and it is now suitable for publication.

Reviewer #3 (Remarks to the Author):

I read through the revised manuscript.

Authors added experiment using full-cells, which provides improved data.

By the additional experiment, dendritic sodium and quasimetallic sodium are observed and adequately analyzed.

All of my queries were solved by the revision.

Reviewers' comments:

Reviewer #1 (Remarks to the Author):

Reviewers' comments:

Reviewer #1 (Remarks to the Author):

The work reported in this manuscript was significantly improved, with new experiments carried out using better electrodes and a full cell configuration. Concerning the analysis of voltage profiles, a few points would still deserve some revision before publication.

The discussion of figures 2 and SI2 remains misleading and partially erroneous. Indeed fig. SI2, especially cycle 1, does not really look like the typical slope-plateau voltage profile for hard C, which would rather look like fig SI10a in an ideal case, and like fig. SI9 in an unoptimized cell. From fig. SI2 it can be seen that the first discharge is stopped after (possibly incomplete) SEI formation, and definitely before the low-voltage plateau is reached. There is no evidence that any sodium remains trapped inside the carbon structure since the ~ 45 mAh g⁻¹ irreversible capacity can entirely be attributed to this SEI formation. On the other hand, it is actually interesting that at cycle 1 the analysis is carried after SEI formation only. As indicated in the description of fig. SI5, no major changes in the distribution of Na species are seen within the cell, and the reason is most probably that no significant Na insertion in the carbon occurred since 30 mAh g⁻¹ reversible capacity corresponds to the very beginning of the slopy region.

Authors response: We agree with the reviewer's comments and have mended the manuscript in the following way:

(black – original text from manuscript; **highlighted**– additional text and amendments; ~~strikethrough~~– removed text)

Page 8: Figure 2 shows ²³Na NMR spectra, 2D images and 1D profiles for a cell which underwent a single discharge/charge (30 mA g⁻¹) cycle outside the magnet. The corresponding potential vs. capacity profile is shown in the supplementary information (Fig. SI2), which exhibits the typical slope-plateau profile characteristic of hard carbons. The capacity is significantly less than that achieved in a traditional coin cell setup which is due to the high resistivity of the cell configuration, **and hence the lower voltage limit is reached**

before full sodiation occurs. This result however does give us a good indication of the different sodium speciation within or on the hard carbon, and also gives us the ability to study the morphology of sodium plating. The first cycle Coulombic efficiency is 10%, indicating that a the majority of the sodium transferred in the first cycle has either gone to form the electrolyte interface on the carbon or remains trapped in the carbon after charging. Due to the limited reversible capacity it is likely that we observed mostly the sodium speciation at the interface in this case. The peak for the metallic sodium (1131 ppm) remains unchanged, but there are now four peaks observed for the sodium in the electrolyte, at chemical shifts of 1, 0, -5 and -7 ppm (Fig. S14 and Table S13). The two peaks at 0 and -5 ppm have chemical shifts, peak intensities and line widths consistent with those observed in the pristine cell, for electrolytic sodium within the separator and excess 'free' electrolyte outside the cell (Fig. 1 and S14). The additional peak at -7 ppm appears in the chemical shift range expected for sodium within the microporous carbon electrode¹⁵, shifted by ring current effects. There are two stages of sodiation in hard carbon, the first follows a sloping voltage profile and is thought to be intercalation between the turbostratic graphene layers, the second is a voltage plateau near to 0 V vs Na/Na+, which is thought to be metal-like sodium precipitating at the vertices of the layers, or pooling of the sodium. However, in this case the full sodiation was not observed and although both mechanisms have been observed to occur simultaneously, the very low capacity and only the sloping voltage profile suggests that in this case it is unlikely that the second sodiation species was observed, and only limited occurrence of the first sodiation species is likely due to the low specific capacity.^{13, 16} The fourth peak, at a chemical shift (1 ppm) nearest to the electrolytic sodium in the separator, has a much broader line width (1900 Hz) compared to that of the peak at 0 ppm (450 Hz).

Page 9: A second charge cycle was performed on the cell, at a specific current of 20 mA g⁻¹ inside the magnet, but no significant changes were observed in the subsequent spectra or images (Figure S15). Again, the data indicate the sodium is inserted into the carbon electrode, rather than plated, and from the electrochemical data (see S1), whilst a similar degree of sodiation is achieved during the second discharge (0.1103 mA•h vs 0.1271 mA•h in the first cycle), more sodium can be extracted when charging at a lower specific current (0.0591 mA•h vs 0.0133 mA•h in the first cycle).

Page 10: The promotion of galvanostatic plating was investigated following discharge at a specific current of 150 mA g⁻¹. Under these conditions, diffusion or intercalation of the sodium through the electrode is kinetically unfavourable and plating is expected to occur preferentially to insertion, which is consistent with the ²³Na NMR spectra, 2D images and 1D profiles shown in Figure 3.

Page 12: The signal at -7 ppm also appears to remain relatively unaffected during this process, as is expected because of the limited amount of sodium inserted into the carbon.

Page 13: However, the formation of quasimetallic species in the carbon was not observed, this is most likely because of the low level of sodium transfer, and the second sodiation species normally observed at the low voltage plateau does not occur, as can be seen in figure S12, where only a quarter of the capacity is observed.

Reviewers comments: The full cell is charged to 4.18 V, where plating is believed to occur. No sodium plating is actually seen in this cell and the authors mentioned in their rebuttal letter that plating is relatively small because of the 30% excess capacity of the cathode. However, it is likely that this capacity excess is compensated by the irreversible capacity at the anode, which is hardly reproducible from one type of cell to another, especially when moving to operando cells. Figure S10c illustrates well the purpose of the authors by assembling a full cell with overcapacitive cathode to reach Na plating on the anode. It also shows that the irreversible capacity is higher in the full cell, with more irreversible capacity on the anode side than in the half cell, since the last inflexion at 0.14 mAh cm⁻² in the C half-cell is seen later in the full cell, around 0.25 mAh cm⁻². This feature is not attributable to the cathode which shows a monotonous voltage trend vs capacity in fig. S10b. More irreversible capacity implies that plating will occur later, at a stage that possibly was not reached in the full cell. If reduction was extended below 0V in the C half-cell, Na plating would occur after a nucleation peak that would be easily identified in the voltage profile of a full cell as well. In absence of such peak during the charge of the cell, it can be believed that plating did actually not occur. Nevertheless, relevant information is obtained from this work regarding the state of Na in hard C at low voltage.

Authors response: The reviewer makes a useful suggestions and we have mended the manuscript in the following way: (black – original text from manuscript; highlighted – additional text and amendments)

Page 16: To investigate sodium plating, and the persistence of the sodium clusters, a new full-cell was charged to 4.18 V, where plating is believed to occur (Fig S10). *In situ* ²³Na NMR spectra were acquired, which are shown in figure 7, immediately after the cell was charged, 4 h after charging and after the cell was discharged. The spectra after charging (Fig 7. a,b) reveal signal in a region of the spectrum (> 400 ppm) where quasimetallic clusters were observed *in operando*, however, no metallic sodium signal is observed. This absence of plating is possibly caused by irreversible capacity within the cell, which utilises the excess sodium during the SEI formation on the hard carbon. These spectra also reveal an evolution of these sodium species, shown by the increase in chemical shift of this peak after 4 h, suggesting an increase in the size of the clusters¹⁶. This quasimetallic peak disappears after the battery is discharged (Fig. 5c). The ²³Na NMR spectra also show new sodium environments in the solvated region of the spectrum following charging, as seen previously, where a distribution of peaks are observed down to –160 ppm. These environments are also seen to evolve over time, where after 4 h these peaks mostly disappeared.

Reviewer #2 (Remarks to the Author):

The authors have addressed the concerns I had with the original version of the manuscript, and it is now suitable for publication.

Reviewer #3 (Remarks to the Author):

I read through the revised manuscript.

Authors added experiment using full-cells, which provides improved data.

By the additional experiment, dendritic sodium and quasimetallic sodium are observed and adequately analyzed.

All of my queries were solved by the revision.

REVIEWERS' COMMENTS:

Reviewer #1 (Remarks to the Author):

I believe this manuscript is suitable for publication in its present form.